

# Stylometry for real-world expert coders: a zero-shot approach

Andrea Gurioli[1], Maurizio Gabbrielli[1] and Stefano Zacchiroli[2]

[1] Department of Computer Science and Engineering, University of Bologna, Bologna, Italy
[2] LTCI, Télécom Paris, Institut Polytechnique de Paris, Paris, France

## ABSTRACT

Code stylometry is the application of stylometry techniques to determine the authorship of software source code snippets. It is used in the industry to address use cases like plagiarism detection, code audits, and code review assignments. Most works in the code stylometry literature use machine learning techniques and (1) rely on datasets coming from *in vitro* coding competition for training, and (2) only attempt to recognize authors present in the training dataset (in-distribution authors). In this work we give a fresh look at code stylometry and challenge both these assumptions: (1) we recognize expert authors who contribute to real-world open-source projects, and (2) we show how to accurately recognize authors not present in the training set (out-distribution authors). We assemble a novel open dataset of code snippets for code stylometry tasks consisting of 114,400 code snippets, authored by 104 authors having contributed 1,100 snippets each. We develop a K-nearest neighbors algorithm (k-NN) classifier for the code stylometry task and train it on the dataset. Our system achieves a top accuracy of 69% among five randomly selected in-distribution authors, thus improving state of the art by more than 20%. We also show that when moving from in-distribution to out-distribution authors, the classification performances of the k-NN classifier remain the same, achieving a top accuracy of 71% among five randomly-selected out-distribution authors.

## INTRODUCTION

*Code stylometry*[1] is the application of stylometry techniques to determine the authorship of software source code snippets (*Oman & Cook, 1989*). Code authorship originates from forensics applications and is an important task in tackling issues such as (de)anonymization and is helpful in various application domains such as forensics techniques and plagiarism (*Oman & Cook, 1989*). In the state of the art, the problem is addressed mainly by using machine learning methodologies (*Bogomolov et al., 2021*; *Kovalenko et al., 2020*), focusing on the training of classifiers, starting from an existing dataset of code examples whose authors are known.

Since the emergence of the concept of author's fingerprint presence in code (*Oman & Cook, 1989*), which emphasized the potential for disambiguating authors based on their programming style, the feature extraction phase for code vectorization has become a pivotal

Corresponding author
Andrea Gurioli,
andrea.gurioli5@unibo.it

[1] Also referred to in the literature as *code authorship attribution* and *code author recognition*.

juncture for recognizing authors, highlighting two distinct approaches. One approach involves leveraging Abstract Syntax Tree (AST) as the primary source of the author's fingerprint representation (*Caliskan-Islam et al., 2015*; *Dauber et al., 2018*; *Hozhabrierdi, Hitos & Mohan, 2020*; *Alsulami et al., 2017*), while the other approach does not (*Oman & Cook, 1989*; *Kurtukova, Romanov & Shelupanov, 2020*), relying solely upon features extracted directly from the source code. Presently, the majority of studies in this field rely on the Google Code Jam dataset (GCJ) (*Caliskan-Islam et al., 2015*; *Bogomolov et al., 2021*; *Alsulami et al., 2017*), which is a collection of data labeled with author information obtained from the algorithm competition of the same name. This choice ensures that the model acquires disambiguation capabilities related to the style of the author rather than the specific problem considered in the code snippet (*Caliskan-Islam et al., 2015*).

Considering this approach, it is possible to develop a tool that operates effectively in an ideal scenario but falls short in a realistic setting. *Dauber et al. (2018)* addressed this issue by mining a dataset from GitHub's repositories and examining how different code characteristics, such as the character count of the input, could influence the model's overall accuracy. In a similar vein, *Kurtukova, Romanov & Shelupanov (2020)* introduced the concept of "expert users", *i.e.,* programmers following coding standards for software development, emphasizing how adhering to specific programming writing patterns—as expert users do—could unify the style of the authors, negatively impacting on the accuracy of stylometric tools.

Several use cases beyond conventional classification methods have emerged with advancements in disambiguation techniques. In particular, the "zero-shot" paradigm (*Chang et al., 2008*) has permitted the possibility of classifying classes not seen during the training phase. Achieving a zero-shot setup for code authorship attribution offers enhanced flexibility, improving the usability of the model while eliminating the need for fine-tuning phases that require experienced users (*Bogdanova, 2021*; *Hozhabrierdi, Hitos & Mohan, 2020*). Currently, the existing models tailored for zero-shot setups for authorship attribution are trained and evaluated using the GCJ dataset (*Hozhabrierdi, Hitos & Mohan, 2020*; *Bogdanova, 2021*). The use of this dataset makes the obtained models not very reliable when it comes to developing an author disambiguation tool for real-world scenarios, as the GCJ dataset is somehow "artificial" and does not reflect the real practice of expert programming.

In light of this, our primary objective in this paper is to obtain an authorship attribution classifier that works well on code snippets from real-world open-source projects. This means that we consider multiple-authored source code projects with authors adhering to established quality-of-code standards, such as those typically used by expert programmers. The approach is influenced by the findings presented in *Dauber et al. (2018)*, which demonstrate an inverse relationship between the length of the code snippet and the ability of the tool to disambiguate authors. Moreover, we are interested both in known authors, that is, authors whose code fragments are known and contained in the available dataset (so-called *in-distribution authors*), and in authors who are not present in the dataset (*out-distribution authors*). Our main research question can then be stated as follows:

- **RQ**: Is it possible to devise a code stylometry technique, based on machine learning, that allows to recognize expert authors of real-world open-source code, with high accuracy for both authors present in the training set (in-distribution authors) and authors not present in it (out-distribution authors)?

## Paper contributions

Our first contribution is the construction of a novel open dataset of code snippets for code stylometry tasks, which was mined from GitHub's public repositories, focusing the authors' lookup on highly skilled developers by using as a seed page search the *libraries.io* ranking. We have partitioned the raw dataset into two main subsets: (1) the *in-distribution* dataset, composed by 104 different authors with 1,100 snippets of code each, used for training and testing; (2) the out-distribution dataset, composed by other 104 authors with 110 snippets each, exclusively used for testing purposes and comprising authors distinct from those in the training set.

As a second contribution, we have then trained and evaluated on our dataset, three AST-base code stylometry classifiers based on *code2seq* and differentiated by attention mechanism and training objective. The evaluation phase yielded positive evidence in response to the research question. Specifically, the self-attention code2seq model (pruned by the classification head and used to generate snippet embeddings for subsequent k-NN classification) achieved accuracies of 69.10% (±10.30) on the in-distribution dataset and 71.40% (±8.40) on the out-distribution dataset, both involving five different "expert" authors, thus improving state of the art by more than 20%.

## Paper structure

The work presented in this paper commences with an in-depth analysis of the existing literature, providing a comprehensive overview of the current state of the art in the field of code stylometry. Subsequently, 'Methodology' delineates the techniques employed, comprising two primary components: the data mining phase and the development of the stylometric model. The obtained results are then presented in 'Results' and discussed in 'Discussion'. 'Conclusions' summarizes the work and also highlights directions for future work.

# RELATED WORK

## Code stylometry with *in vitro* datasets

The work developed by *Oman & Cook (1989)* pioneered the concept of identifying the human fingerprint in code. By leveraging human-driven classification, that work demonstrated the possibility of attributing source code to its author by identifying common writing patterns present in the code. This experiment influenced the adoption of cluster-based classification, introducing an unsupervised technique for inferring the code author. Notably, a key distinction between the work of *Oman & Cook (1989)* and succeeding studies lies in the absence of AST features in *Oman & Cook (1989)*, while these represent an important aspect in current state-of-the-art models.

*Caliskan-Islam et al. (2015)* made significant contributions to the field of authorship attribution by incorporating AST features and using snippet vectorization with syntactical

features derived from AST. This work addressed the authorship problem from various perspectives and demonstrated that syntactical features are less susceptible to obfuscation processes; thus, their use results in a more reliable model. Both syntactical and lexical features (stream of tokens in the source code) were used in this work, together with a random forest classifier. Moreover, *Caliskan-Islam et al. (2015)* were the first to use the GCJ dataset for the training procedure.

With the emergence of word embeddings (*Mikolov et al., 2013*), new techniques for code representation have evolved, which exploited advanced vectorization approaches such as code2vec (*Alon et al., 2019*) in addition to previous methods like LSTM and RF (*Alsulami et al., 2017*; *Bogomolov et al., 2021*; *Kovalenko et al., 2020*). These advancements have achieved remarkable accuracies up to 95.90% (*Bogomolov et al., 2021*) on the GCJ dataset with 70 different authors.

Recently, by adopting a novel methodology, *Hozhabrierdi, Hitos & Mohan (2020)* approached the problem by using embeddings and cosine similarities in the classification process. This approach enabled the exploration of zero-shot authorship scenarios, leading to new research areas. Various training objectives were employed, including metric learning techniques such as triplet loss and cross-entropy losses. These techniques yielded distinct representations in the latent space, with cross-entropy trained models (*Horiguchi, Ikami & Aizawa, 2020*) demonstrating superior performance, particularly when coupled with a k-NN classifier.

Our work leverages AST-based models, working with embeddings in order to obtain zero-shot capabilities. However, we use a different approach from the works discussed above since we avoid "*in vitro*" datasets that contain programs—developed by different authors—that address the same problems. We use a newly defined dataset mined from the public repositories of GitHub, thus obtaining reliable outcomes when classifying authors from real-world case scenarios.

## From GCJ to the real world

The work presented by *Dauber et al. (2018)* brought about a significant paradigm shift by transitioning from the GCJ dataset to a real-world use-case scenario, wherein data was extracted directly from GitHub. This study examined the relationship between model accuracy and snippet size, revealing an inverse correlation between these two variables. Moreover, shifting from an "*in vitro*" corpus, based on algorithmic competitions, to a real-world scenario resulted in a noticeable decrease in overall accuracy, dropping from 53.91% with 229 authors (*Caliskan-Islam et al., 2015*) to 48.80% with 104 authors (*Dauber et al., 2018*). This finding showed the increased difficulty of the stylometry task in real-world applications and emphasized the importance of focusing on performance metrics relevant to practical use cases.

*Kurtukova, Romanov & Shelupanov (2020)* extended the analysis to real-world use cases and examined the impact of various factors on the model's performance. Specifically, they investigated the influence of the author's experience, snippet length, obfuscation processes, and different programming languages. Furthermore, they introduced the concept of an "expert" author, an experienced programmer adhering to specific coding standards

and styles that facilitate collaboration with other developers. Interestingly, their findings revealed that the performance of the model was negatively affected by programming experience: while an average accuracy of 95% was achieved when recognizing randomly inexperienced authors with different coding styles, the accuracy dropped below 50% when inferring "expert" authors from snippets of code up to 1,000 characters per file (*Kurtukova, Romanov & Shelupanov, 2020*). This result highlights the challenges associated with accurately identifying authors who employ standards and code guidelines, highlighting how the standardization process leads to less variability and prominent unification in terms of coding style fading the "fingerprint" of the author, clearly indicating the need for robust models capable of handling such scenarios.

Our work targets explicitly "expert" authors, shifting the focus from a single project authorship task (*Kurtukova, Romanov & Shelupanov, 2020*) to authors belonging to several real-world open-source projects and analyzing short segments of code (less than 1,000 characters per file). We employ distinct models and classification techniques w.r.t. *Kurtukova, Romanov & Shelupanov (2020)* and *Dauber et al. (2018)*, and we emphasize the identification of authors who are absent from the training data (out-distribution authors). Our work will be thus compared with *Kurtukova, Romanov & Shelupanov (2020)*'s work over five "expert" authors (keeping therefore the same number of authors for comparison), leveraging methodologies that enable out-distribution capabilities (k-NN classifier), observing if these techniques can lead to effective and more flexible outcomes without accuracy degradation. Our results will also be compared with *Dauber et al. (2018)* for the whole in-distribution dataset (104 different authors) with both k-NN classifier and architectures with classification head, taking into account the possible increase of difficulty in the disambiguation process given by the dataset design (authors experience and snippet length).

## METHODOLOGY

In the following, we describe the corpus used during the training and testing phases ('Dataset construction'), including details on its mining process, as well as designs and training techniques employed for the code stylometry model ('Stylometric models').

### Dataset construction

Collecting a robust corpus is crucial in order to achieve a high-performing model for our task. *Caliskan-Islam et al. (2015)* provided some indications on how the corpus for code stylometry related tasks should be designed. Here we follow their approach, thus our first focus is to recognise the need to have a wide range of projects per author, for the data mining phase. As discussed later, this approach allows us to train a model capable of author disambiguation based on coding style rather than project-specific characteristics. Our second focus is identifying and representing experienced authors who follow specific coding standards. To achieve this, we need first to mine a set of seed projects in terms of good quality of code and the presence of software engineering practices (as defined, for example, in *Munaiah et al. (2017)*, thus we consider projects that have good documentation, testing, and project management. To maximize the stylometric model's

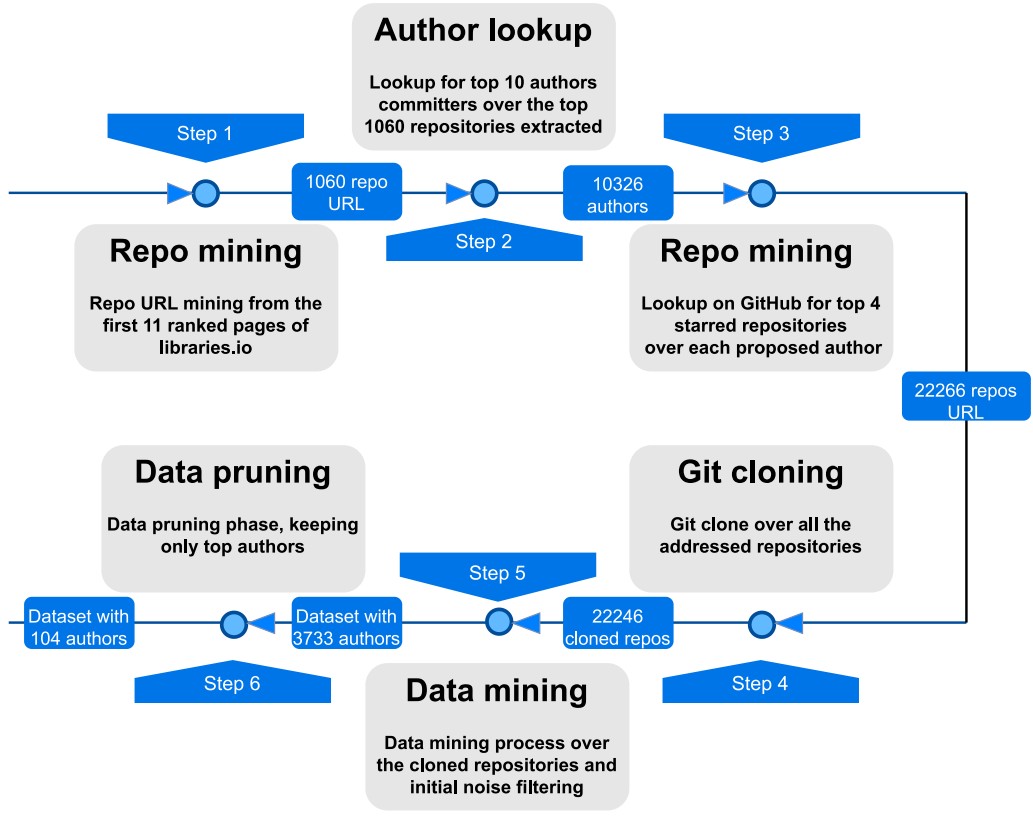

**Figure 1    Data gathering process executed to assemble the experimental dataset.**

usability and flexibility, we represent the source code of large projects as fragments rather than complete files, attributing each fragment to its respective author. Thus, we recognize the unique contributions made by individual authors within the project.

Considering the objectives mentioned above, we divide the data mining phase into the six steps shown in Fig. 1. Note that we will use several hyperparameters that are defined in this section and will be instantiated in 'Dataset'. The process begins with an initial Repository mining, which involves creating a seed list comprising high-quality repositories. This seed list is generated by extracting the results from the top-ranked (*i.e., P = 11*) pages obtained from *Libraries.io (2022)* extracted from PyPI. We apply an initial filtering step, focusing only on the GitHub hosting service. Thus, we ease the subsequent mining processes (such as repositories per author lookup) using only one API service. We filter the list, focusing the lookup on repositories containing Python source code. As a second step, we proceed by mining the set of the most collaborative authors (with *A = 10*, as a maximum amount of different authors threshold) based on the number of commits, representing our population.

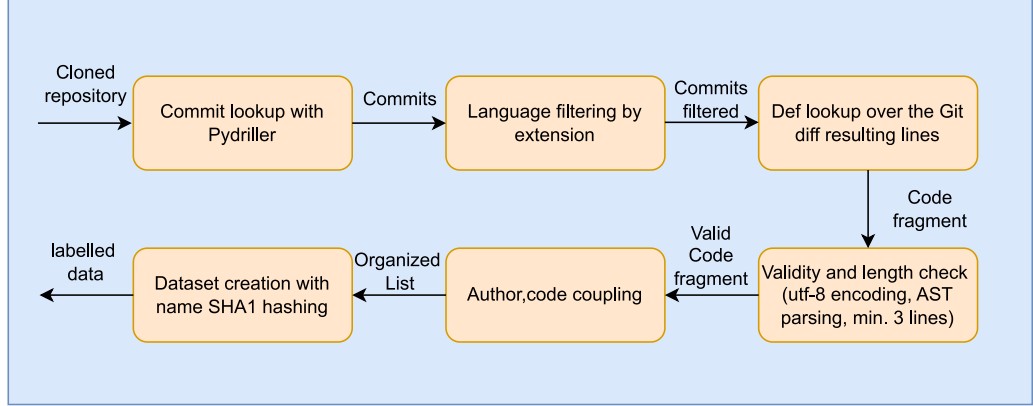

**Figure 2** Data mining process used to extract code snippets from code repositories (detailed view of Step 5 in Fig. 1).

Given the initial list of repositories and seed authors, the third step allows us to extend the initial list of repositories. We proceed thus by mining at most $R = 4$ repositories, sorted by using GitHub stars, for each author and joining the mined list to the initial seed one.

In the fourth step, we clone the repositories for the following data mining process, keeping track of URLs and timestamps, obtaining a local source code archive and a filtered updated list of the valid previously mined repositories.

Figure 2 zooms into the fifth step, which highlights the key phases of the data mining process. The main goal of this step is to extract code fragments associated with a single author, ensuring an accurate representation of the author's "fingerprint". We traversed in temporal order the commits using Pydriller (*Spadini, Aniche & Bacchelli, 2018*), filtered files by language (Python), and extracted the respective git diff. In order to obtain a dataset centered on code snippets, each source code was divided by looking for function definitions using regular expressions with the *def* keyword as a primary criterion.

For each def statement, the algorithm moves incrementally through subsequent lines of code, added in the commit (Fig. 3, "Git diff outcome") and addressable to one author through the meta-data related to the commit, by appending each line to a temporary list of code lines addressed to the current lookup. The algorithm sets the definition as terminated when another definition statement occurs or the further git diff added lines are not contiguous, thus joining the temporary list to a unique string by examining the validity of the code's snippet for parseability to verify syntactical correctness. The fragments of code related to a commit, after checking whether the overall length is greater than or equal to three lines of code, are aggregated into a list and finally linked with their respective author (taken from the meta-data related to the commit), thereby obtaining the corresponding labels (Fig. 3, 'Unique author functioning listing'). In the final phase of the fifth stage, these labeled fragments are stored locally. To ensure uniqueness within the dataset, the *SHA1* hashing algorithm is utilized to generate filenames based on the content of each code snippet. This prevents the occurrence of duplicates in the dataset (Fig. 3, 'labeled data').

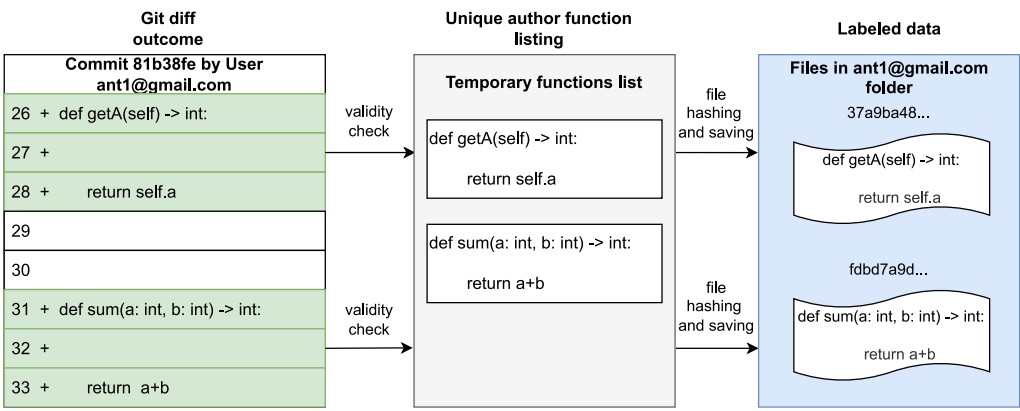

**Figure 3** Extraction of code snippets from "diffs" recorded by code repositories.

We then apply an initial noise reduction before the RAW dataset creation, keeping only authors with at least $IN = 100$ code snippets.

In the sixth step, we conducted a "top authors" selection, wherein developers with fewer than $F = 1,100$ code fragments are pruned from the dataset. However, we retain a RAW version of the dataset for further analysis. The sixth step is repeated on the RAW dataset to create a second dataset specifically for out-distribution tests. This is achieved by filtering the RAW dataset to extract authors with an overall snippet count lower than the previously mentioned threshold $F = 1,100$ but higher than $FZ = 800$.

## Stylometric models
### Model design

Due to their superior accuracy performance in stylometric tasks (*Alsulami et al., 2017*; *Bogomolov et al., 2021*), AST-based models have been widely adopted. In our study, we have chosen the code2seq (*Alon, Levy & Yahav, 2018*) snippet embedding structure as a common base for developing three different designs differentiated primarily by means of the attention technique, learning objective and the technique used to handle the vocabulary. We initially discuss the common part (Fig. 4) of the three architectures, an advancement of the *Alon et al. (2019)* model that encodes snippets by extracting stream tokens (leaves) and their ancestors from the Abstract Syntax Tree.

The pre-processing needed to extract the AST from the input code snippets is conducted using the Tree Sitter library (*TreeSitter, 2022*): This provides a parser generator tool capable of retrieving a representation of the snippet's Concrete Syntax Tree which will be then simplified (*e.g.,* by pruning redundant syntactical elements as brackets or punctuation nodes) to an Abstract Syntax Tree. In such an AST each node corresponds to one or more nodes in the concrete syntax tree as follows: the leaves correspond to the "stream tokens" (*i.e.,* terminal nodes) while their ancestors are the non-terminal nodes. Following *Alon et al. (2019)* we define the AST as follows:

**Definition 1 (Abstract Syntax Tree (AST))** *An Abstract Syntax Tree (AST) for a code snippet* $C$ *is a tuple* $\langle N, T, X, s, \delta, \phi \rangle$ *where:* $N$ *is a set of non-terminal nodes,* $T$ *is a set of*

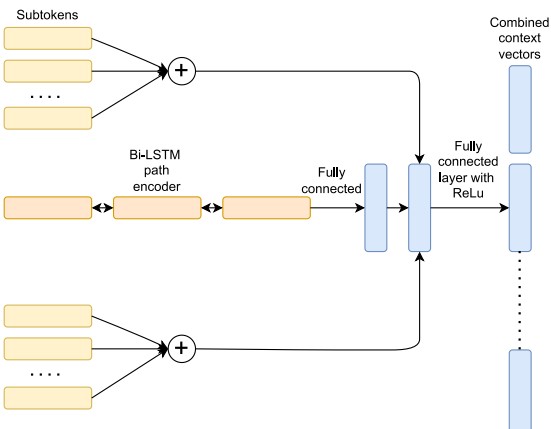

**Figure 4 Common architecture of the three proposed models.**

*terminal nodes, $X$ is a set of values, $s \in N$ is the root node, $\delta : N \rightarrow (N \cup T)^*$ is a function that maps a non-terminal node to a list of its children, $\phi : T \rightarrow X$ is a function that maps a terminal node to an associated value. Every node, except the AST root, appears exactly once in all the lists of children.*

The architecture uses, as a primary source of input, triplets of elements extracted from the AST, composed by random pairs of terminal nodes which will be split into subtokens, and their channeling path, an ordered list of the terminal nodes' ancestors (non-terminal nodes) extracted by ascending the AST until finding the lowest common ancestor.

To split the terminal nodes into subtokens, we introduce a mixed usage of regular expression (splitting the terminal nodes' tokens by means of camel notation or underscores, *e.g.*, from `arraySplitter` to `array`, `Splitter`), which was already adopted in the original code2seq design, and a subsequent lower-cased SentencePiece encoding (see *Kudo & Richardson (2018)*, pruned by the space inclusive underscore character in order to maintain a stream token representation in line with *Alon, Levy & Yahav (2018)*. In this paper, we introduce a novel use of this technique in order to try to overcome the open-vocabulary problem caused by variable names, thus gaining capabilities of unknown and rare words encoding and therefore using more effectively the code2seq subtoken architecture, which sums the subtokens embedding representation.

Having the aim to disambiguate authors by means of coding style, we decided to strip comments into a common '*comment*' token that emphasizes only the decision of the author of commenting on the code, eliminating thus biases in the inference phase given by the potential presence of Natural language.

We now discuss how the architectures that we use in this work handle the previously defined AST representation of code snippets. Given as input a predefined set of triples $L = \{L_1, \ldots, L_n\}$, let us consider the $i$th path where $X_{sTi} \in \mathbb{R}^{|X_{sTi}| \times d} = [t_1, \ldots, t_m]$ is the starting sequence of subtokens (with an embedding dimensionality $d$ which defines the number of parameters used), while and $X_{eTi} \in \mathbb{R}^{|X_{eTi}| \times d}$ is the ending sequence of subtokens. The subtokens are encoded with an element-wise summation in the embedded representation

to obtain the following:

$$T_{Si} = \sum_{t \in X_{sTi}} t$$

$$T_{Ei} = \sum_{t \in X_{eTi}} t$$

The AST path input, corresponding to a set of non-terminal tokens defined as $X_{ASTi} \in \mathbb{R}^{|X_{ASTi}| \times d}$, is encoded by means of a bidirectional LSTM as follows:

$$h_{1i}, \ldots, h_{li} = LSTM(X_{AST1i}, \ldots, X_{ASTli})$$

where $h_{li}$ corresponds to the $l$th LSTM's hidden state representation of the $i$th triple.

We denote by

$$ASTrepr_i = [\overrightarrow{h_{li}}, \overleftarrow{h_{1i}}]$$

where $ASTrepr \in \mathbb{R}^{1 \times 2d}$, the result of the concatenation of the final hidden states of the bidirectional LSTM.

To combine the triplet in a context vector, we first use a $W_{ast} \in \mathbb{R}^{2d \times d}$ projection matrix to reduce the dimensionality of $ASTrepr$ to the dimension $d$, thus obtaining:

$$ASTred_i = ASTrepr_i \cdot W_{ast}.$$

The next step is then mapping the concatenated encoding of the two stream tokens and the AST representation $[T_{Si}, ASTred, T_{Ei}]$ to the context vector. In order to achieve this we first reduce the dimension by using the (multiplication by the) $W_{ctx} \in \mathbb{R}^{3d \times d}$ context projection matrix and then we use the ReLU activation function:

$$ContextVector_i = ReLU([T_{Si}, ASTred, T_{Ei}] \cdot W_{ctx})$$

It is worth noting that differently from the original code2seq and code2vec architectures (*Alon, Levy & Yahav, 2018*; *Alon et al., 2019*), we have chosen to use the ReLU activation function rather than the tanh, maintaining the tanh function exclusively in the soft attention based classification model. This choice is motivated by the fact that using the tanh activation function we experienced a much longer time for the convergence of the model in the training procedure.

So far, we have described the part of the architecture that is common to the three models we have considered. Next, we describe the parts that differentiate the models, namely the attention mechanism, the learning objective, the dropout presence, and the use of the SentencePiece technique.

The model named as **softAttn-classifier** (Fig. 5) relies on the SentencePiece technique with a compressed vocabulary of 64,000 different tokens for the terminal nodes and non-terminal nodes related vocabulary of 174 different tokens (without any term of compression). The absence of the SentencePiece technique on the non-terminal nodes pre-processing phase is common to the three architectures. This is due to the strict closed-vocabulary use case, which implies the absence of out-of-vocabulary tokens (OoV). This model uses the soft-attention mechanism introduced by code2vec (*Alon et al., 2019*),

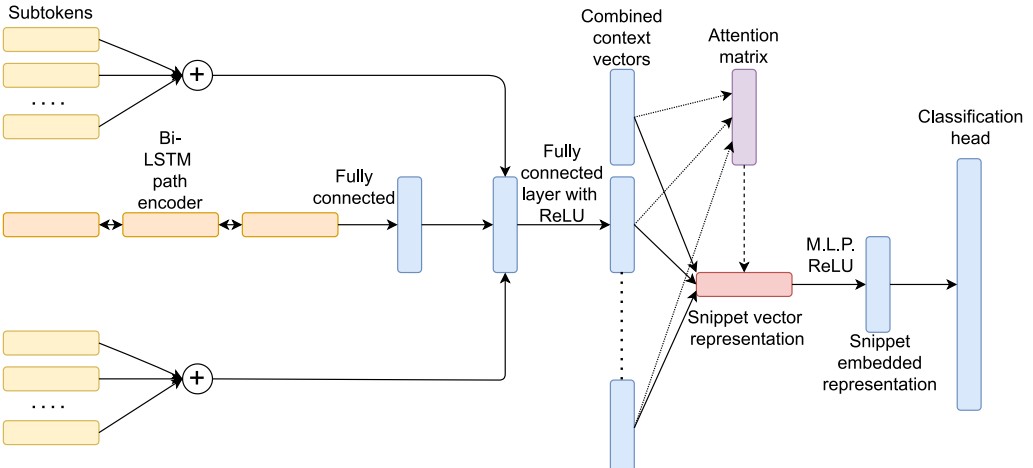

**Figure 5** Machine learning model with soft attention mechanism.

creating thus a hybrid version between the advancements of code2seq in terms of sparseness-related issues with the code2vec-shaped output. After the attention mechanism, we have applied an M.L.P. layer with a ReLU activation function ($512 \rightarrow$ ReLU $\rightarrow 512 \rightarrow$ ReLU $\rightarrow 256$) gaining non-linearity advantages (*Appalaraju et al., 2020*) for the final vectorial representation, useful for a k-NN classification (see 'Classification process'). We then have a final classification head with $A = 104$ nodes as the overall number of authors obtained from the in-distribution dataset in the former mining process. We have conducted the training process with the Cross-Entropy loss. The model has respectively an embedding (previously referred to as $d$) and LSTM hidden representation of 256 units. We have then exploited Dropout as a data regularization factor (0.25 right after the context embedding concatenation).

The second model, referred to as **selfAttn-classifier** (Fig. 6) shares the same vocabulary compression algorithm as the previously depicted model for the terminal nodes, having the SentencePiece technique with a compressed vocabulary of 64,000 tokens and a non-terminal nodes related vocabulary of 174 different tokens. This model replaces the soft attention technique with a self-attention mechanism (transformer architecture (*Vaswani et al., 2017*) inspired by the design used in the work of *Radford et al. (2021)*. Since the code2seq model utilizes random pairs, no positional encoding and no autoregressive masking are added, preserving the core random representation concept of code2seq by leveraging the positional invariance properties of the architecture. The output of the transformer is subsequently normalized with a Layer normalization technique, flattened, re-projected to a lower dimensionality of 512 units using a ReLU activation function, and then passed to the classification head (with $A = 104$ nodes). We have conducted the training process with a Cross-Entropy loss. The model consists of an embedding (previously referred to as $d$) and Bi-LSTM hidden representation of 512 units, along with a transformer architecture

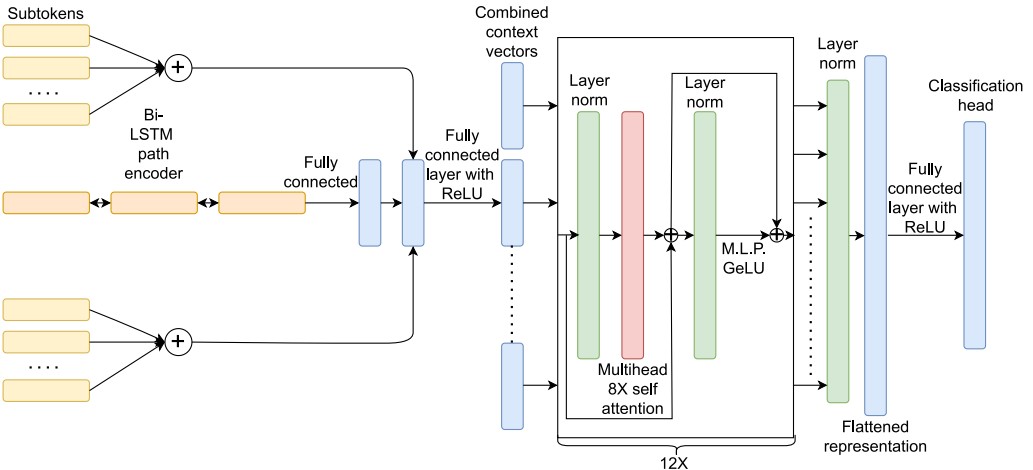

**Figure 6  Machine learning model with self-attention mechanism.**

comprising a 12-layer, 512-wide model with 8 attention heads, as described by *Radford et al. (2021)*.

The third model, named **ML model** (like Fig. 5, but without the final classification head), as we experienced better performances, does not rely on the SentencePiece algorithm, having thus a vocabulary of 147,967 tokens for the terminal nodes and, like the preceding architectures, encompasses 174 different values for the non-terminal ones. This model relies on the soft-attention mechanism as *Alon et al. (2019)*. After the attention mechanism, we have then applied an M.L.P. layer with a ReLU activation function (512 → ReLU → 512 → ReLU → 256) gaining non-linearity advantages (*Appalaraju et al., 2020*). The training process, in this case, is conducted with the infoNCE loss (further training details on 'Training') introduced by *van den Oord, Li & Vinyals (2018)*. The model relies on an embedding $d$ and LSTM hidden representation of 256 units and uses a dropout regularization of 0.25 after the context embedding concatenation.

### Training

We trained the first two classification models (softAttn-classifier and selfAttn-classifier) using a stochastic gradient descent optimizer with a learning rate scheduler. We set the initial learning rate to 0.01, and the scheduler followed a step size of 50 epochs with a gamma value of 0.95. Each training batch had a dimension of 64 elements. The softAttn-classifier model incorporates input boundaries, following the approach described by *Bogomolov et al. (2021)*. Specifically, we used AST boundaries of four units regarding the distance between leaves and a maximum path length (composed of non-terminal tokens) of seven elements. The two models were trained for 1200 epochs, and the best checkpoint, determined by the best results in terms of validation loss, was saved and used for testing purposes.

The ML model, trained with the infoNCE loss proposed by *van den Oord, Li & Vinyals (2018)*, modified by *Zhang et al. (2022)* operates on $N$ pairs of distinct snippets from the same class. We obtain batches of dimensions $N^2$, where the matrix diagonal of the batch

**Figure 7** Representation of the batch-making process.

represents the ground truth over the snippets representations on the latent space $(v_i, u_i)$ as the i-th pair (Fig. 7). To measure the distance between embeddings, we employ the **cosine similarity** with the L2 norm:

$$\langle v_i, u_i \rangle = \frac{v_i^T u_i}{||v_i|| \, ||u_i||}$$

We then apply the cosine similarity in the following loss function:

$$\ell_i^{(v \to u)} = -\log \frac{\exp(\langle v_i, u_i \rangle / \tau)}{\sum_{k=1}^{N} \exp(\langle v_i, u_k \rangle / \tau)}$$

$$\ell_i^{(u \to v)} = -\log \frac{\exp(\langle u_i, v_i \rangle / \tau)}{\sum_{k=1}^{N} \exp(\langle u_i, v_k \rangle / \tau)}$$

$$Loss = \frac{1}{N} \sum_{i=1}^{N} (\ell_i^{(v \to u)} + \ell_i^{(u \to v)})/2$$

The final loss is calculated as the mean of the two previously derived losses, following the approach established by *Radford et al. (2021)*. The parameter $\tau$ denotes the temperature parameter that governs the range of logits in the softmax function; we fixed it at 0.1.

Given that the batch determines the negative and positive samples, it is imperative to customize the sampler to ensure that only distinct classes are selected to prevent the divergence in the latent space of elements belonging to the same class. Consequently, this introduces a limitation on batch dimensionality during training, where the total number of classes present in the dataset determines the maximum batch size.

We have trained the metric learning-based model (ML model) for 1,200 epochs with Adam optimizer, an initial learning rate of 0.01, applying a scheduler with a step size of 50 epochs and a gamma value of 0.95. Each training batch had a dimension of $64^2$ elements. As the softAttn-classifier model, we incorporated input boundaries, using AST boundaries

of 4 units regarding the distance between leaves and a maximum path length of 7 elements. As the former models, we kept the best checkpoint in terms of validation loss.

### Classification process

We utilized the latent space representations obtained from the code snippets for the classification task to calculate similarities using a k-NN classifier. We employed the cosine similarity metric to measure similarity, aligning with the ML model for consistency. This choice of similarity metric, as highlighted by *Horiguchi, Ikami & Aizawa (2020)*, resulted in improved embedding representations even for the cross-entropy trained models.

To obtain the latent space representation, we removed the classification head from both the softAttn-classifier and selfAttn-classifier models, resulting in a dimensionality of 256 for the softAttn-classifier model and 512 for the selfAttn-classifier model. The ML model retained a dimensionality of 256 units. We have settled the k-NN classifier to work as a 1-NN classifier, using the closest representation as the inferred outcome. We have processed the data for the NN classifier by projecting in the latent space and computing similarities of each snippet of code from the testing set when it comes to evaluating the in-distribution properties, and zero-shot test set for the out-distribution properties; discarding then the first result as it would always be the querying snippet representation. To provide a basis for comparison with previous work by *Dauber et al. (2018)* and *Kurtukova, Romanov & Shelupanov (2020)*, we have conducted the experiments with author subsets consisting of five, 10, 20, and 104 authors. To ensure randomness, a random author picker was employed. We have performed five experiment iterations, and the results were reported with the mean and standard deviation, considering the significant source of variance introduced by the random author selection.

## RESULTS

We present below our main results: the novel curated dataset for code stylometry of expert coders in 'Dataset' and the different code authorship attribution models in 'Stylometric models'.

### Dataset

The raw dataset we consider is the outcome of the first six phases described in 'Dataset construction' and contains snippets of code labeled by author emails as unique identifiers. We instantiated the hyperparameters with respectively $P = 11$ ranked pages for the first step—obtaining 1,060 repositories' URLs–and $A = 10$ different authors per project for the second step—thus forming an authors' list of 10,326 different entities. As for the third step, we have set $R = 4$ repositories per author, finalizing the repositories' URL list with a total amount of 22,266 elements. In the cloning phase (phase four), we have cloned the whole repositories' URL list, thus producing 22,246 repositories stored locally. The difference in terms of number of elements between the third and fourth phases is attributed to URLs that are either invalid or no longer accessible.

The noise reduction in the fifth step is settled by taking only authors with $IN = 100$ code snippets, thus obtaining a RAW dataset of 3,733 authors. The RAW dataset is then pruned

**Table 1  Dataset dimensionality after the splitting and under-sampling phases.**

| Subset | Authors | Dataset share (%) | Fragments per author |
|---|---|---|---|
| Training (in-distribution) | 104 | 80 | 880 |
| Validation (in-distribution) | 104 | 10 | 110 |
| Test (in-distribution) | 104 | 10 | 110 |
| Zero-shot (out-distribution) | 104 | NA | 110 |

**Table 2  Analysis of mean and median character's (symbols) values in the datasets.**

| Subset | Mean symbols | Median symbols |
|---|---|---|
| Training-val-test (in-distribution) | 785 | 301 |
| test set (in-distribution) | 566 | 298 |
| Zero-shot (out-distribution) | 600 | 281 |

**Table 3  1-NN accuracies over the test set dataset (in-distribution).** The best results are highlighted in bold.

| Model | 5 authors (%) | 10 authors (%) | 20 authors (%) | 104 authors (%) |
|---|---|---|---|---|
| softAttn-classifier | 67.2 (±8.9) | **61.8** (±3.3) | 49.60 (±6.0) | 35.2 (±0.1) |
| selfAttn-classifier | **69.1** (±10.3) | 52.0 (±6.7) | 43.4 (±6.0) | 31.6 (±0.1) |
| ML model | 62.0 (±7.8) | 61.6 (±6.0) | **51.0** (±6.7) | **35.5** (±0.1) |

by all the authors having less than $F = 1,100$ code snippets for the in-distribution set. As for the out-distribution zero-shot set, we took from the RAW dataset authors having between $FZ = 800$ and $F = 1,100$ (1,100 excluded) code snippets. This allows us to obtain a list of authors with a high level of variance in terms of data while making sure that authors do not coexist in the two datasets.

After the pruning phase, authors from both the in-distribution and out-distribution sets are randomly picked, thus obtaining 104 overall different classes. The resulting sets are then randomly under-sampled to 1,100 snippets for the in-distribution set, which will be split into train-set, val-set, and test set, respectively, with 880, 110 and 110 snippets of code (Table 1). We also obtain 110 snippets for the out-distribution test set. We then tested the mean and the median values for the number of symbols. Given the inverse correlation between the snippet length and model performances, as shown in Table 2, we obtain two main datasets, which, according to the results in *Dauber et al. (2018)* and *Kurtukova, Romanov & Shelupanov (2020)* could be defined as difficult to infer.

## Stylometric models

All experiments were conducted using the PyTorch framework (*Paszke et al., 2019*), utilizing the NVIDIA A100-40GB graphics card for the training process. The validation set was exclusively used during training to select the best model checkpoint based on validation loss. The results presented below were obtained using the testing sets from the in-distribution (Table 3) and out-distribution (Table 4) datasets. Visual representations shown below (Fig. 8) are obtained using the t-SNE algorithm (*Maaten & Hinton, 2008*).

**Table 4  1-NN accuracies over the Zero-shot dataset (out-distribution).** The best results are highlighted in bold.

| Model | 5 authors (%) | 10 authors (%) | 20 authors (%) | 104 authors (%) |
|---|---|---|---|---|
| softAttn-classifier | 65.6 (±5.6) | 54.5 (±4.6) | **46.4** (±2.9) | 31.6 (±0.1) |
| **selfAttn-classifier** | **71.4** (±8.4) | **59.1** (±6.9) | 45.8 (±4.9) | **36.9** (±0.1) |
| ML model | 51.4 (±5.2) | 44.2 (±1.8) | 34.2 (±1.3) | 22.9 (±0.1) |

The performances of the stylometric models reported in Tables 3 and 4 were calculated in terms of overall accuracy, defined as the ratio between the number of correct classified snippets (most similar snippet belonging to the same class of the snippet to classify) with respect to the total number of snippets to classify. The classification process is obtained by computing the cosine similarity ('Training') between all the snippets' vectorized representations present in the test set. Each snippet is thus classified as the label of the most similar snippet in the latent space.

We report the softAttn-classifier and selfAttn-classifier performances with their respective classification heads (Table 5), which outputs a probability distribution over the number of existing classes, taking the most probable class as the output class. We define the accuracy for the classification-headed models as the ratio between the overall amount of correctly classified snippets with respect to the total number of snippets to classify.

Table 3 presents the outcomes obtained from the in-distribution data, revealing an inverse correlation between the number of authors and the model performances. In comparison to the findings of *Kurtukova, Romanov & Shelupanov (2020)*, our work demonstrates a substantial improvement, with a general top accuracy of 69.1% (selfAttn-classifier) for five distinct authors. In contrast, Kurtukova's accuracy is lower than 50%. It is important to note that this comparison is based on the "expert" authors' results from their work, which utilized files with less than 1,000 symbols, whereas our study focused on expert users with a test-set median of 298 symbols and a mean of 785 symbols (in the in-distribution dataset), examining thus a dataset which, according to the results of *Kurtukova, Romanov & Shelupanov (2020)*, is potentially more *difficult* to classify.

Moreover, the performance deterioration ($-3.6\%$) with the 1-NN classifier for 104 in-distribution different authors between the selfAttn-classifier and the softAttn-classifier highlights that better outcomes with the classification head do not imply better embedding representations.

The out-distribution results presented in Table 4 demonstrate the superior representation capabilities of the selfAttn-classifier model. While the in-distribution results did not reveal a clear performance advantage among different architectures, the out-distribution results indicate a significant negative disparity (*e.g.*, $-8.7\%$ for the 104 authors between softAttn-classifier and the ML model with 104 different authors) between the ML model and the models trained with the classification head. This discrepancy could be attributed to the absence of the SentencePiece technique in the ML model. This technique offers the classification head-trained models enhanced word embedding representations for unknown out-of-vocabulary words and could positively impact out-distribution

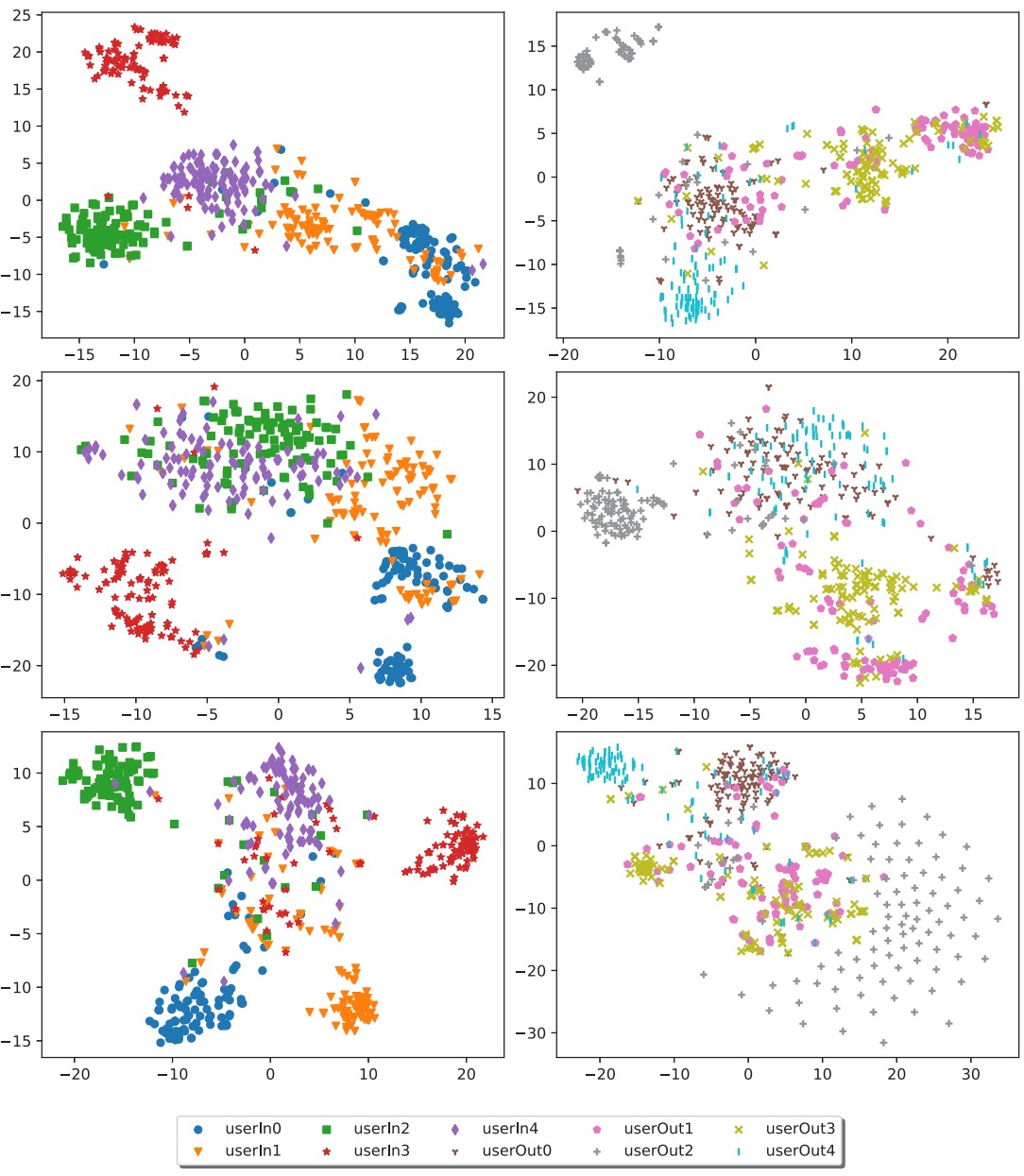

**Figure 8** **Embeddings latent space representations for five different in-distribution authors (depicted in the legend as *userIn*) and five different out-distribution authors (*userOut*).** From top to bottom, the embeddings are derived from the softAttn-classifier, selfAttn-classifier, and ML model. In the left column, we display embeddings for the in-distribution authors, while the right column showcases embeddings for out-distribution authors.

performance. These results emphasize the necessity of conducting an ablation study in future work.

It is worth noting that comparing the results of the in-distribution and out-distribution selfAttn-classifier model, the out-distribution results did not compromise its representation

**Table 5   Classification accuracies for the in-distribution dataset for the two classification-trained models.** The best results are highlighted in bold.

| Model | 104 authors (%) |
|---|---|
| softAttn-classifier | 42.5 (±0.1) |
| **selfAttn-classifier** | **44.8** (±0.1) |

**Table 6   Time of convergence compared to models' dimensionality.**

| Model | # Parameters | Time per epoch (s) | Time convergence (hours) |
|---|---|---|---|
| softAttn-classifier | ≈18M | 245.1 (±4.8) | ≈76 |
| selfAttn-classifier | ≈114M | 1526.4 (±37.3) | ≈295 |

properties, obtaining even better performances. This observation suggests that self-attention mechanisms could imply greater resilience and generalization properties to unknown authors' recognition (out-distribution data).

The accuracies achieved using the classification head are shown in Table 5, demonstrating higher performance compared to the 1-NN classifiers. However, it is important to note that the standard setup, which utilizes the classification head, lacks flexibility and cannot be used for zero-shot classification tasks.

In addition to basic accuracy measures, we conducted a comparative analysis of the two *softAttn-classifier* and *selfAttn-classifier* models (Table 6) to gain deeper insights regarding models' dimension and training time. Despite exhibiting superior performance, the selfAttn-classifier relies on an architecture 6 times larger, resulting in slower training times and higher memory consumption due to the scaling number of parameters. However, it proved to be more effective overall as the selfAttn-classifier model reached the softAttn-classifier best performances (44.8% in terms of accuracy with the classification head) in 21 epochs (≈ 9 h of training time). In contrast, the softAttn-classifier model reached these performances in 1,080 epochs (≈ 73 h of training time).

## DISCUSSION

Based on the experimental outcomes, we can positively answer the stated research question: it is possible to devise a code stylometry machine learning technique that recognizes expert authors of real-world open-source code with remarkable accuracy compared to the results of *Kurtukova, Romanov & Shelupanov (2020)*, keeping consistent outcomes between in-distribution and out-distribution authors.

By considering their style, we created a model that disambiguates "expert authors" of real-world open-source projects. We reached a high level of accuracy on both in-distribution and out-distribution authors. Our model outperformed the results of previous models (*Kurtukova, Romanov & Shelupanov, 2020*) by 20% for both in-distribution and out-distribution authors (results obtained with the selfAttn-classifier model over five different authors).

When considering the results in *Dauber et al. (2018)* we obtain worse performances: −13.3% with 1-NN classifier ML model and −4% with the classification headed selfAttn-classifier model, both on 104 different authors. However, the results obtained by *Kurtukova, Romanov & Shelupanov (2020)* suggest that this degradation can be attributed to the fact that *Dauber et al. (2018)* considered inexperienced authors rather than expert authors as we do. We can state that worse accuracies regarding models tested on experienced authors could be attributed to a faded fingerprint induced by following coding standards and guidelines, thus obtaining a unified coding style.

The absence of performance degradation observed when transitioning from in-distribution to out-distribution authors underscores the robustness of the models in fulfilling out-distribution tasks, thereby confirming an essential aspect of our research question about their high accuracy regarding out-distribution authors.

This property is mainly present with the selfAttn-classifier model, highlighting how a self-attention-based architecture can be crucial to achieving this flexibility. As *Caliskan-Islam et al. (2015)* stated, having a dataset with few projects per author can lead to models disambiguating authors by means of the project and not by coding style; obtaining models resilient to unknown authors (with zero-shot capabilities) emphasizes thus the quality of the dataset in terms of different projects per author.

In Fig. 8, we have visualized author snippets over the latent space, thus showing how different training techniques lead to different representations. In the in-distribution representation, the ML model trained with the infoNCE loss displays greater confidence in tightening and driving away authors' clusters. In the out-distribution results, superior snippets representation can be spotted by considering the classification-based models, following the outcomes of Table 4. All the visual results show the presence of hard recognizable snippets that can be blamed on the presence of highly commonly used coding standards, possibly leading to a less identifiable fingerprint of the author.

We also introduced two main novelties in the code2seq architecture: the usage of the SentencePiece technique and the model's output representation. Since the original code2seq model was designed to output a sequence, we developed a hybrid architecture between code2vec and code2seq, introducing two different attention mechanisms. The selfAttn-classifier was capable of capturing the style of the author without boundaries, effectively, achieving better performances than the bounded model (+2.3% in classification setup compared to softAttn-classifier). This improvement could be due to the self-attention capabilities that allow for effective representation of long-term dependencies between paths. However, we need to note that our dataset is based on fragmented source codes, which probably avoids the overfitting problem described by *Bogomolov et al. (2021)*. An ablation study should be conducted to address the strengths and weaknesses of these models more precisely.

The new dataset obtained from the mining process designed by us tackles the data sparseness problem indicated by *Alon et al. (2019)*. It enables us to obtain a training set with 880 unique snippets of code for 104 different authors, thus increasing the data volume obtained by *Dauber et al. (2018)* from 150 to 1,100 snippets of code per author. In comparison, the work of *Kurtukova, Romanov & Shelupanov (2020)* on expert authors

relies on a training set of 30 files per author, constraining the usage to models that do not suffer from data hungriness.

It is also worth emphasizing that our dataset differs from the one mined by *Kurtukova, Romanov & Shelupanov (2020)*, depicting a domain shift in terms of the considered "experience" of the authors to obtain a broader impact on the code authorship task. *Kurtukova, Romanov & Shelupanov (2020)* defines the "expert" author as one who follows coding standards focusing on a single open-source project (Linux kernel). Our work instead considers authors from several different projects, defining expertise as the feature deriving from following coding guidelines when producing open-source code. Our work addressed this change of perspective in considering "expertise" by developing a model capable of disambiguating authors through different projects despite similarity elements that can deteriorate the specific fingerprint of the author.

Furthermore, the code snippets are labeled using the addressed author in the commit. As in the commit procedure, the author can be revised or encompass multiple hidden collaborators, which could lead to a weak representation of the developers' style who contributed to the code. We can highlight this as a possible noise source for our stylometric models, which could cause incorrect disambiguation. Thus, we outline an open-source based dataset that is structurally more challenging to disambiguate than an "in vitro" dataset. Aiming to comprehend these difficulties, we can initially address them to both the "expertise" of the authors that homologates the coding style to a similar fingerprint and the possibility of having mislabeled snippets of code, leaving room for further investigations.

## CONCLUSIONS

We have investigated whether it is possible to achieve good performance on code stylometry for "expert" coders contributing to real-world open-source projects. We leverage techniques suitable for recognizing both in-distribution and out-distribution authors, allowing use independent of the authors seen in the learning phase. We have obtained remarkable results (20% better accuracy with respect to *Kurtukova, Romanov & Shelupanov (2020)*, for both in- and out-distribution authors) on a novel open dataset of experienced authors and short code snippets authored by them.

We have furthermore introduced a hybrid architecture between AST-based models and the transformer design. Our architecture shows how self-attention algorithms, aided by a scaled model, can improve accuracy performances (+2.2% in classification setup) and do not degrade from in-distribution to out-distribution inference.

### Future work

We plan to conduct an ablation study to examine the model dimensionality and the presence of SentencePiece and bounded inputs. We expect this to provide evidence-based explanations for some of the observed strengths and weaknesses of the models we have developed.

We have considered all output tokens from the transformer as feature representations as we have an arbitrary pre-determined number of input triplets. This choice results in a scaled number of parameters over the first layer of the final M.L.P. layer. The model

could work more efficiently by introducing a special token ((CLS) classification token) in the vocabulary and using such token as the sole source of feature representation for the subsequent M.L.P. layer as in *Devlin et al. (2018)*.

It is also worth noting that the architectures described rely on randomly selected input triplets, avoiding any form of positional information. We want to further explore this, as the incorporation of positionality into the input source code could potentially serve as a valuable source of characterization regarding the author's style, impacting the performances of the models.

Finally, exploring the resilience of these models to code obfuscation techniques represents an intriguing area for further investigation.

## Data availability

A complete replication package for this work is available for download from Zenodo at https://doi.org/10.5281/zenodo.10796494 in which datasets and checkpoints can also be found.

### Funding

The authors received no funding for this work.

### Competing Interests

The authors declare there are no competing interests.

### Author Contributions

- Andrea Gurioli conceived and designed the experiments, performed the experiments, analyzed the data, performed the computation work, prepared figures and/or tables, authored or reviewed drafts of the article, and approved the final draft.
- Maurizio Gabbrielli conceived and designed the experiments, analyzed the data, prepared figures and/or tables, authored or reviewed drafts of the article, and approved the final draft.
- Stefano Zacchiroli conceived and designed the experiments, analyzed the data, prepared figures and/or tables, authored or reviewed drafts of the article, and approved the final draft.

### Data Availability

A complete replication package for this work is available at Zenodo: Gurioli, A., Gabbrielli, M., & Zacchiroli, S. (2024). Stylometry for Real-World Expert Coders: a Zero-shot Approach - Replication Package. https://doi.org/10.5281/zenodo.10796494.

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
