# Peer review of "Stylometry for real-world expert coders: a zero-shot approach"

_PeerJ Computer Science, doi:10.7717/peerj-cs.2429_

## Round 0.1 · accepted · Accept

All of the reviewers are enthusiastic about the manuscript already in its current form. You have clear research questions and a suitable study design. You provide a data set and show interesting conclusions.

The only comments that the reviewers had, I trust you to incorporate them in the camera-ready version:

- Refer to the paper from Dauber et al. https://doi.org/10.1145/3183440.3195007
- Check the proposed changes in how you formulated some sentences.

Reviewer 1 ·

Basic reporting

The article is written in clear and unambiguous professional English .
Literature references re sufficient
Article structure , figures, tables are professional.
Self content results

Experimental design

The aim and scope of the research work are clearly defined, relevant, and meaningful.
Through investigation performed to high technical and ethical standards.
Methodology is explained in detail.

Validity of the findings

An open dataset of code snippets for code stylometry is created. Impact and novelty were accessed. Conclusions are well stated.

Cite this review as

Reviewer 2 ·

Basic reporting

The paper is well structured, and the information is presented in a straightforward manner. Informative figures and detailed descriptions of the processes further enhance the clarity of the presentation.
I commend the authors for their meticulous approach and consideration of various aspects of the problem at hand. Their attention to detail and in-depth understanding of the subject matter are evident.

Regarding language use, the paper is well-written, with very few issues where I would suggest (optional) rephrasing:

1) RQ, line 71-72: I would propose rephrasing with “allows the recognition of expert authors…”
2) throughout the paper: "out-of-distribution" instead of "out-distribution"
3) line 397: "of overall accuracy, defined as the ratio between the number of correctly classified snippets"
4) Line 525: "special token ([CLS] classification token) in the vocabulary and using such a token/such tokens as the sole source..."

Some abbreviations are not properly introduced the first time they are used:

Line 114: LSTM, RF
Line 290: M.L.P.

Experimental design

No comment

Validity of the findings

No comment.

Cite this review as

·

Basic reporting

The manuscript uses professional English and provides sufficient context on the topic of code stylometry. It references relevant literature and aligns with the journal's standards.
However, the text uses passive voice frequently, which can be revised to an active voice for better engagement. For example: "a k-NN classifier was developed" could be "we developed a k-NN classifier."


Figures and tables are clear, well-labeled, and relevant to the study, offering a good visual explanation of the data.

Experimental design

The research question is meaningful and aims to address a gap by focusing on expert coders contributing to real-world projects, moving away from datasets that originate from coding competitions. However, referring to the dataset as novel is farfetched as https://faculty.washington.edu/aylin/papers/gitBlameWho.pdf already explores the use of github code fragments for code stylometry

The authors present a robust experimental setup, with detailed steps on data mining and model training.

The k-NN classifier is well-explained, and the results are compared against state-of-the-art models. However, further explanation of the model limitations is necessary.

Validity of the findings

The study is statistically sound and supports its conclusions with data. It compares models trained with and without classification heads, demonstrating improved performance with their self-attention-based classifier.
The zero-shot results, especially for out-of-distribution authors, provide new insights. Nevertheless, the impact of some limitations, such as the influence of coding standards that reduce variability between expert authors, should be explored further.

Additional comments

The paper tackles an important problem in code stylometry and presents a well-designed method that could enhance the field.

Cite this review as